# Neural efficiency in EFL learning and positive psychology

**Liwei Hsu** [ID]*

Graduate Institute of Hospitality Management, National Kaohsiung University of Hospitality and Tourism, Kaohsiung City, Taiwan

* liweihsu@mail.nkuht.edu.tw

## Abstract

This study aims to investigate how teachers' positive feedback influences learners' positive psychology in English as a Foreign Language (EFL) instruction and to examine how positive psychology enhances learning outcomes through the neural efficiency hypothesis. The study employed three neuroscientific and behavioral experiments to address these questions. Study 1, with 78 college-level EFL learners, examined the effect of teachers' positive feedback on learners' positive psychology and found a significant positive impact. Study 2, conducted with 102 EFL learners, tested the neural efficiency hypothesis and revealed that higher positive psychology improved neural efficiency, with positive feedback moderating this effect. Study 3, involving 54 EFL learners, explored the relationship between neural efficiency and learning outcomes, confirming that enhanced neural efficiency leads to better learning performance. Beyond education, the results underscore the broader relevance of positive psychology and neural efficiency in other domains, such as workplace performance, skill acquisition, and personal development, where maintaining cognitive efficiency and well-being is crucial. These findings highlight the importance of positive feedback in fostering positive psychology, improving neural efficiency, and enhancing learning outcomes. The study offers theoretical contributions and practical recommendations for EFL educators to leverage positive feedback and promote effective learning.

## Introduction

The processes involved in language learning and teaching are complex and multifaceted, as they intertwine cognitive, affective, and social factors that shape the acquisition of linguistic knowledge [1]. Among these, affective factors are critical in foreign language learning [2, 3]. Positive psychology (PP), a psychological framework pioneered by [4], seeks to enhance subjective well-being and improve quality of life by focusing on concepts that promote individual happiness and psychological health [5]. Since 2014, research has increasingly explored the relationship between PP and language learning, though the field remains in its formative stages, with further empirical validation needed [6]. The integration of PP into language learning and teaching has opened new avenues of inquiry [7], driving a paradigm shift toward enhancing learners' experiences—particularly in second language (L2) acquisition [2, 8]. Scholars argue

**Data Availability Statement:** The data are uploaded as supplementary information.

**Funding:** This study was financially support by the National Science and Technology Council (111-2410-H-328 -004 -MY2). The funders had no role in study design, data collection and analysis,

decision to publish, or preparation of the manuscript.

**Competing interests:** The author has declared that no competing interests exist.

that incorporating PP offers a transformative lens for language education [9]. Over the past decade, studies investigating L2 acquisition through PP have increased significantly, highlighting its potential to revolutionize pedagogical practices, boost learner engagement, and improve learning outcomes [10–12]. However, existing studies have primarily focused on the general benefits of PP in fostering motivation and well-being [13], leaving gaps in our understanding of how these psychological mechanisms operate on a cognitive-neural level to enhance language learning outcomes. Additionally, many studies have emphasized subjective emotional experiences but have not systematically investigated the physiological processes underlying these benefits, such as the role of neural efficiency.

This study addresses these gaps by adopting a neuroscientific approach to examine the impact of teachers' positive feedback on learners' PP and its connection to improved cognitive functioning in second language (L2) acquisition. Specifically, we apply the *neural efficiency hypothesis* [14] to explain how positive psychological states enhance cognitive performance through optimized neural resource allocation. While prior research (e.g., [15]) has identified connections between emotion, cognition, and neural responses, the current study offers new insights by showing how positive feedback interacts with learners' affective states to foster neural efficiency, which, in turn, leads to better language learning outcomes. This present research advances the field by going beyond behavioral observations to integrate neurophysiological measures, providing a deeper understanding of the cognitive processes that underlie the role of PP in language learning. This focus on neural mechanisms marks a significant departure from earlier studies, which have primarily examined PP at the behavioral or emotional level without directly linking it to brain efficiency or learning performance.

In language learning, feedback is critical in shaping learners' progress, motivation, and engagement [16]. While the effects of corrective feedback have been extensively studied, the significance of positive feedback remains underexplored, particularly within the context of EFL education [17]. This research seeks to address this gap by examining the impact of positive feedback on EFL learners' PP, mental well-being, self-confidence, and learning outcomes. Furthermore, the study highlights the practical advantages of adopting positive feedback strategies in EFL classrooms by analyzing empirical evidence, theoretical frameworks, and pedagogical implications.

The overall objective of this study is to examine how teachers' positive feedback influences PP in EFL learners and investigate how PP enhances learning outcomes using the neural efficiency hypothesis. Specifically, it aims to answer the following questions:

1. Does teachers' positive feedback improve EFL learners' PP?

2. How does EFL learners' PP affect their neural efficiency?

3. Does improved neural efficiency lead to better learning outcomes?

The novelty of this research lies in its experimental design, using frontal alpha asymmetry (FAA) of EEG to objectively quantify the impact of positive feedback (compared to no feedback and criticism) on learners' PP. The findings provide valuable theoretical insights and practical implications for educators across diverse educational contexts beyond Taiwan. The study underscores the importance of creating supportive and encouraging classroom environments by demonstrating that positive feedback enhances learners' PP, neural efficiency, and learning outcomes. Educators in multicultural or multilingual settings can leverage positive feedback strategies to boost learners' confidence, motivation, and engagement, which is particularly crucial for students facing challenges such as language anxiety or low self-efficacy. Furthermore, incorporating positive reinforcement into teaching practices can also foster

resilience and well-being in learners, which is essential in both formal and non-formal education settings worldwide.

## Literature review and hypotheses development

### Positive psychology and EFL learning

PP emphasizes the positive dimensions of human experience, such as happiness, strengths, and life satisfaction, contrasting with conventional psychology, which often addresses mental health issues [18]. Foundational theories within PP, such as *broaden-and-build theory* of [19] and *control-value theory* of [20], have shaped second language acquisition (SLA) research since 2012 (Li, 2020). Specifically, PP examines psycho-emotional constructs, including happiness, life satisfaction, optimism, gratitude, and resilience [4, 21].

In the context of SLA, PP facilitates learners' ability to identify personal strengths and overcome challenges [22]. Empirical studies suggest that learners' engagement with positive psychological factors significantly affects their motivation and academic success. For instance, a study of [23] found that fostering a positive classroom environment and promoting a growth mindset increase students' willingness to communicate in English, particularly when supported by teachers. Similarly, a survey conducted by [15] of Chinese EFL learners demonstrated that enjoyment of English learning correlates with higher self-confidence and improved self-assessments of proficiency, findings echoed by [24]. [3] further reported that PP interventions effectively reduced anxiety among EFL learners over time.

PP enhances linguistic outcomes and cultivates non-linguistic competencies, such as intercultural awareness and general learning abilities [6]. [2] highlighted the value of balancing positive and negative emotions in language learning, showing that learners who practiced emotion regulation across three experimental trials with 209 EFL students experienced improved motivation, reduced anxiety, and enhanced proficiency. However, despite these advances, the neural mechanisms underlying PP in language learning remain unexplored. This study seeks to investigate these mechanisms and their influence on learning outcomes.

Various surveys and instruments assess PP in educational contexts [25]. PP interventions are typically designed to foster positive emotions, including happiness and well-being [26]. Electroencephalogram (EEG) research has linked positive emotional traits to frontal EEG asymmetry, particularly frontal alpha asymmetry (FAA), which has been implicated in emotional well-being [27]. Corroborating this, [28, 29] identified significant correlations between FAA and subjective well-being, building on the findings of [30]. [31] also employed the FAA to assess improvements in medical students' well-being following positive psychological interventions. Further exploration of the relationship between FAA and positive emotions represents a promising avenue for future research [32–34].

### The importance of teachers' positive feedback for EFL learners' PP

[35] define "feedback" as information provided by teachers or peers based on evaluations of one's performance on specific tasks. Effective teacher feedback is crucial in improving student achievement [36], with high-information feedback—particularly when coupled with self-regulation strategies—proving especially beneficial [37, 38].

Feedback is essential for EFL learners' proficiency in language acquisition. [39] emphasized its importance for developing English writing skills, while [40] demonstrated that computer-based feedback enhances learners' writing self-efficacy. [16] further observed that positive feedback improves EFL learning outcomes, even when conveyed through emojis. Additionally, teacher support significantly influences learners' positive psychological states [41–43].

The interplay between teachers' positive feedback and learners' PP shapes both language acquisition and the creation of a supportive academic environment. Understanding these dynamics allows teachers to empower learners both academically and emotionally. Developing feedback-rich environments can effectively support students in navigating language learning challenges. However, the mechanisms through which teachers' positive feedback enhances EFL learners' PP remain an open question for further investigation. Thus, this study proposes the following hypothesis to be examined:

*H1*: *Receiving teachers' positive feedback significantly increases EFL learners' PP in learning English.*

## The neural efficiency hypothesis in language learning

The neural efficiency hypothesis posits that individuals with higher intelligence exhibit reduced brain activation during cognitive tasks, reflecting greater efficiency [44]. Neural efficiency is shaped by the quantity and quality of learning experiences [14]. [45] further proposed that language aptitude results from an advantageous neurocognitive profile, fostering high intrinsic motivation and proactive engagement in language learning. Their research highlights the importance of specific brain regions and networks, such as the bilateral auditory cortex and auditory-motor connections, in determining language learning success. Faster, more concise activation of these areas reflects enhanced neural efficiency [46].

Neural efficiency has also been consistently observed among proficient native language speakers [47]. [48] found that multilingualism is associated with greater neural efficiency in native language production. Similarly, [49] suggested that neural efficiency supports the development of neurocognitive strategies that leverage brain processes to facilitate language learning, improving vocabulary recall and recognition in English learners. It is hypothesized that EFL learners' neural efficiency can improve through real-time feedback, which may help optimize neural patterns for more effective language processing [50]. The following hypotheses were proposed:

*H2*: *EFL learners' neural efficiency in EFL learning is significantly higher when having a higher level of PP with the teacher's positive feedback.*

*H2a*: *EFL learners' neural efficiency in EFL learning is significantly higher when receiving positive feedback from the teacher.*

*H2b*: *EFL learners' neural efficiency in EFL learning is significantly higher when having a higher level of PP.*

*H3*: *EFL learners' learning outcomes are significantly associated with their level of neural efficiency.*

The use of EEG has become increasingly common in assessing neural efficiency, enabling researchers to explore the optimal allocation of cognitive resources across various tasks [51, 52]. Research indicates that alpha oscillations (8–13 Hz) are prominent in human EEG and play a key role in cognitive processes related to neural efficiency. Specifically, alpha oscillations help inhibit irrelevant neural activity, facilitating efficient processing of relevant information [52]. As a result, EEG has been proposed as a reliable tool for measuring neural efficiency during learning [53]. For instance, tracking changes in alpha power during problem-solving tasks can reveal individual differences in how cognitive resources are allocated [54]. Building on this framework, the present study will employ EEG to assess participants' neural efficiency.

**Table 1. Overview of the studies and findings.**

| Studies | Objective | Design | Main findings |
|---|---|---|---|
| Study 1 | Test the hypothesis that English as a Foreign Language (EFL) learners' PP could be improved with positive feedback from the teacher. | Single-factor between-subjects design (with teacher's positive feedback vs. teachers' criticism vs. no feedback). | The teacher's positive feedback enhanced EFL learners' PP |
| Study 2 | Examine how PP affects EFL learners' neural efficiency in learning through the teacher's positive feedback. | Within-subjects 3 (with the teacher's positive feedback vs. with teachers' criticism vs. without feedback) × 2 (high PP vs. low PP) factorial experiment | The participants' PP significantly affected their neural efficiency as well as the teacher's positive feedback, which could successfully enhance their PP. |
| Study 3 | Understand the association between EFL learners' neural efficiency and their learning outcomes. | Single-factor between-subjects design (IV: neural efficiency; DV: learning outcome) | The relationship between neural efficiency and learning outcomes was significant and robust. |

Note: Neural efficiency refers to how well the brain uses energy to process information. A more efficient brain uses less energy to do the same work, leading to better thinking, learning, and memory. Positive psychology (PP) includes what makes people thrive and flourish, such as positive emotions and character traits.

## Overview of this research

Three experiments (Studies 1 to 3) were conducted to test the proposed research hypotheses and investigate their significance systematically. Study 1 explores whether EFL learners' PP can be improved with positive feedback from the teacher (H1). Study 2 examines how EFL learners' PP affects their neural efficiency in learning when the teacher's positive feedback is presented (H2). Study 3 aims to understand the association between EFL learners' neural efficiency and their learning outcomes (H3). Please refer to Table 1 below for an overview and the main findings of this research.

The experiments were conducted at the Sensory Marketing and Mind Science Laboratory of a public university in southern Taiwan between March 1, 2023, and April 5, 2024. As recommended by [55], participants were selected using specific inclusion and exclusion criteria to ensure the reliability and consistency of the findings.

The inclusion criteria for the study was: 1) age between 20 and 25 years; and 2) no history of neurological disorders or traumatic brain injuries. The exclusion criteria were: 1) a history of substance abuse or addiction; and 2) current use of medications that may impact neurological function, such as antipsychotics, antiepileptics, or other neuroactive drugs.

The participants' English proficiency was determined based on their TOEIC scores. Individuals with scores ranging from 500 to 650, or equivalent, were included in the study to enhance the generalizability of the findings, as the average TOEIC score in Taiwan was reported to be 568 in 2021. This range ensures that the participants represent typical intermediate-level English learners, which is suitable for evaluating language processing and proficiency. The designed tasks mainly focused on fundamental areas where learners with TOEIC scores between 500 and 650 typically need improvement, such as vocabulary expansion, sentence structure, and comprehension of academic texts.

Participants came to the laboratory individually. Before the experiment, the procedure details were explained to the participants. All participants provided written informed consent prior to participation. All experimental procedures strictly adhered to the ethical standards outlined in the Declaration of Helsinki and were approved by the Ethical Committee for Human Research of the university in southern Taiwan (NCKU-HREC-E-110-577-2).

## Apparatus and measurement

While increasing the number of channels for signal data collection may lead to more accurate results, it can also diminish the practicality and usefulness of the experiments, ultimately

impacting the real-world applicability of the research findings [56]. Hence, the neurophysiological data for this study were collected using the Procomp Infinity encoder developed by Thought Technology, a system previously employed in academic studies (e.g., [57]). The power spectrum density (PSD) of the EEG data was acquired using an eight-channel Procomp Infinity encoder system from Thought Technology, Inc., employing the International 10–20 system with Cz as the reference and the area between Fz and Fpz as the ground. All electrode impedances were under five kΩ, and a digital average mastoid reference (M1+M2)/2 was used. The electrodes of EEG were placed at specific positions within the 10/20 system: Fp1, Fp2, Fz, F3, F4, F7, F8, and Cz due to their relevance to foreign language acquisition according to [58].

The PSD was calculated from the total power in the alpha band (8–13 Hz), which was then extracted and averaged over minutes. Power values were logarithmically transformed (ln) to obtain normalized data. Additionally, a measure of FAA, extracted using ln [right]–ln [left], revealed that high FAA indicated positive emotions [59, 60]. Furthermore, a more significant reduction in alpha power during cognitive tasks may suggest more efficient neural processing [14]; lower alpha band power can represent greater neural efficiency.

## Course contents

All experiments in this current study were carried out in an online learning context, focusing on the ability of English reading comprehension. The study materials were obtained with permission from a Youtuber @teacherfish1019 (https://www.youtube.com/@teacherfish1019), who assists EFL learners aiming for success in the TOEFL exam through practical strategies that have consistently helped learners achieve their desired TOEFL scores (see Figs 1 to 3 for

**Fig 1. Snapshot of the TOEFL reading instruction 1.**

**Fig 2. Snapshot of the TOEFL reading instruction 2.**

the contents of instruction). Participants in three groups were directed to watch the same video (two units of TOEFL Reading Section) for approximately 30 minutes during the experiments while their EEG brainwaves were recorded. Additionally, through training for the TOEFL reading comprehension examination, participants could develop essential English reading skills such as skimming and scanning [61], critical reading [62], annotating [63], identifying main ideas and supporting details [64], and synthesizing information [65]. Reading passages are chosen carefully to align with participants' current abilities, with initial texts drawn from general-interest sources and simplified academic content. Over time, the texts become longer and denser, reflecting TOEFL's requirements for handling academic reading material. This approach helps to reduce the influence of varying proficiency levels, as noted by [66].

Potential cofounding effects might exist because some learners may perform better on reading or listening tasks related to familiar topics. To mitigate this, the course materials cover a broad range of subjects from humanities, sciences, and social studies, ensuring that all participants are equally challenged. Moreover, proficiency development is influenced by individual motivation and physical well-being. The course schedule is designed to balance intensity and rest, with alternating task types to sustain engagement and prevent burnout.

## Study 1

Study 1, inspired by [31] design, examined the neural correlates of a PP intervention with 78 college-level EFL learners using EEG. The participants' PP was linked to increased FAA

**Fig 3. Snapshot of the TOEFL reading instruction 3.**

associated with adaptive emotion regulation. This study adopted a similar method to investigate whether EFL learners' PP could be improved with positive feedback from the teacher. Study 1 was conducted between April and June of 2023.

## Participants

Study 1 employed a single-factor between-subjects design, with the independent variable being the presence of the teacher's positive feedback (with positive feedback vs. with criticism as the control feedback vs. without feedback) and the dependent variable being the level of participants' PP (an indicator of FAA). The sample size was determined using G*Power 3.1.9.7, with effect size = .40, statistical power = .80, and alpha level = .05 for three independent means. The recommended minimum number of participants was 64. As a result, Study 1 invited 78 participants (26 in the 2 control and 1 experimental groups) from two universities in southern Taiwan ($n = 78$, $M_{age} = 21.5$, $SD_{age} = 1.6$, Male = 32, Female = 46). They were recruited through major social media sites such as Facebook and Line, which college students in Taiwan use. After the participants had agreed to join this experiment, they were randomly assigned to one of the three groups.

## Research procedure

After the participants were assigned to the experimental and control groups, the experimental group received positive feedback from the teacher. In contrast, the control group received criticism or no feedback. The criticism in this research indicated that the instructor often pointed

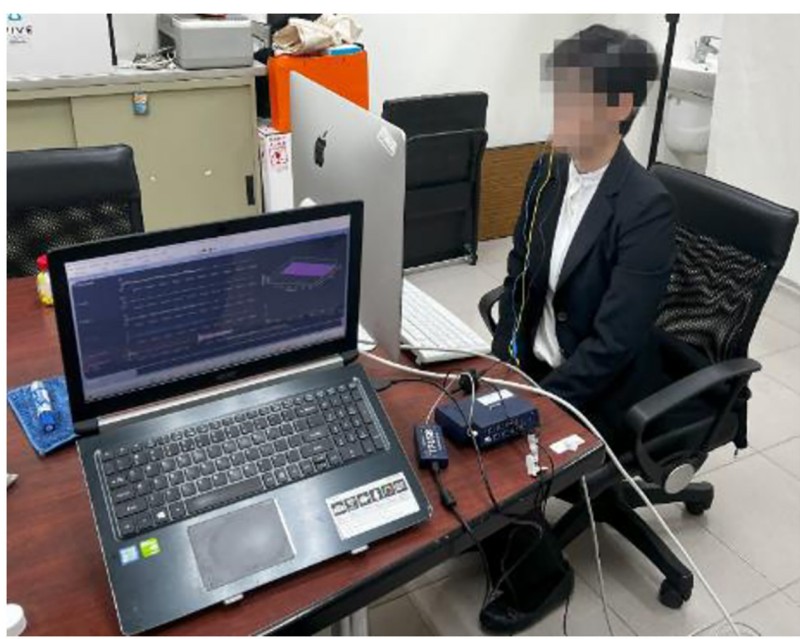

**Fig 4. Research environment and the administration of Study 1.**

out errors while neglecting to acknowledge correct answers. The feedback focused solely on language issues rather than the learners themselves. Students from Confucian-influenced cultures typically experience this type of feedback, as teachers in countries like China tend to provide minimal praise [39]. As a result, the emotional well-being of participants in the control group—who received either criticism or no feedback—can be assured.

When the instruction started, the participants were directed to attach the EEG electrodes and sit with their eyes closed for 2 minutes. Calming music was played to relax the participants, and the collected EEG data served as a baseline reference for the experimental settings. Subsequently, the participants were instructed to sit individually in front of a computer monitor 1 meter away (Please refer to Fig 4 for the representation of Study 1). The online instruction began once they were prepared. Throughout the duration, the participants were instructed to minimize unnecessary movements to reduce the likelihood of generating EEG noise. In the study, the instructors positioned themselves alongside participants to monitor EEG signals and administer feedback based on the research protocol. In the experimental group, positive feedback such as 'good job' and 'great work' was provided when participants completed reading a passage and answered questions correctly. Even when incorrect answers were chosen, participants received encouragement such as 'Do not worry about it' or 'This one is particularly challenging, and you are doing well.' Conversely, participants in the control group did not receive positive feedback, even for correct responses. Instead, they either received criticism regarding their performance, such as 'This question is not particularly difficult, yet the answer was incorrect,' or no feedback.

## Statistical analysis

The analysis of covariance (ANCOVA), which is employed to compare different groups while adjusting for a covariate, was used to analyze the collected data, with participants' baseline FAA serving as the covariate. However, ANCOVA assumptions, including the homogeneity of

the regression slopes, must be assessed given that the incorporation of an interaction term between the intervention and the covariate in the ANCOVA model enables the investigation of ATI (Aptitude-Treatment Interaction) effects [67]. The detection of heterogeneity in regression slopes can reveal ATI effects. Therefore, alternative ANCOVA methods must be explored. In such cases, utilizing statistical formulae from the Johnson-Neyman procedure to establish concurrent regions of significance provides a straightforward alternative [68]. ANCOVA of Study 1 and the statistical analyses of Study 2 and 3 were performed with SPSS version 26.

## Results and discussion of Study 1

The heterogeneity of the regression slopes indicated that the interaction was insignificant ($p$ = .181). Moreover, skewness and kurtosis were computed to evaluate the data distribution's normality. The skewness registered at 0.992, fitting within the normality criteria of -2 to +2. Meanwhile, the kurtosis measured at 0.172, aligning with acceptable norms of -7 to +7. These findings suggest that the data distribution closely resembles normality. Put together, it was sound to use ANCOVA for statistical analysis in Study 1. The findings from the ANCOVA (see Table 2) indicated statistically significant differences ($F_{(2)}$ = 22.952, $p < .001$, $\eta_p^2$ = .383) in the participants' post-intervention FAA (i.e., the dependent variable) between the control and experimental groups when baseline FAA was adjusted for as the covariate. As for the effect size of ANCOVA, $\eta_p^2$ is used to determine the proportion of variance in a dependent variable explained by an independent variable after accounting for other variables in the model. The value of $\eta_p^2$ of Study 1 indicated a large effect size, which is a more thorough verification of the impact uncovered by the ANOVA test [69]. Thus, the participants who received the teacher's positive feedback had significantly higher FAA than those in the control group (Positive Feedback = 1.404, No Feedback = .938, criticism = .896 see Fig 5), indicating that the teacher's positive feedback enhanced EFL learners' PP and supporting H1.

The research findings aligned with previous studies, highlighting the importance of attention, motivation, and emotional factors in predicting positive feedback from teachers. For instance, [36] suggested that positive feedback enhances motivation and engagement in EFL learners; when students receive positive feedback emphasizing their progress and effort, it motivates them to continue their language learning journey. The impact of teacher's positive feedback on EFL learners' PP is substantial, as it motivates them, improves self-regulation, promotes autonomy, and influences goal-oriented behavior. These results highlight the significance of deliberate positive feedback practices in language education. However, it is crucial to acknowledge that Study 1 was an innovative investigation that utilized neuroscientific methods to quantify the participants' PP. Only a few studies have documented statistically significant alterations in EEG signals linked to behavioral evaluations. The underlying mechanism behind these changes is still unknown. Therefore, Study 2 was carried out to delve deeper into this matter using the neural efficiency hypothesis as a framework.

**Table 2. Results of ANCOVA.**

| Cases | Sum of Squares | df | Mean Square | F | p | $\eta_p^2$ |
|---|---|---|---|---|---|---|
| BaselineFAA | 17.221 | 1 | 17.221 | 244.857 | < .001 | 0.768 |
| Group | 3.228 | 2 | 1.614 | 22.952 | < .001 | 0.383 |
| Residuals | 5.205 | 74 | 0.070 | | | |

*Note*. Type III Sum of Squares

Dependent variable: FAA

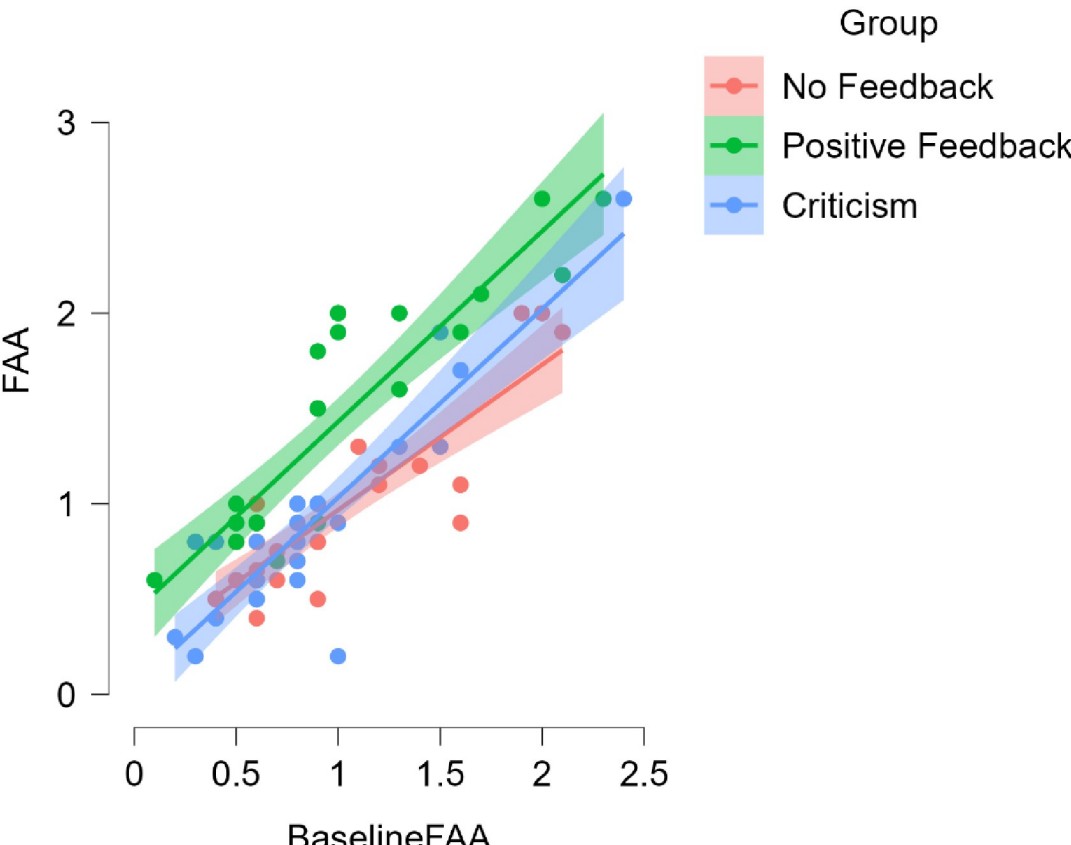

**Fig 5. The effect of the teacher's feedback on the participants' FAA adjusted for baseline FAA.**

## Study 2

Study 2 added one critical dependent variable: the participants' neural efficiency. To better understand how PP affects EFL learners' neural efficiency in learning through the teacher's positive feedback, Study 2 designed a between-subjects 3 (with the teacher's positive feedback vs. with the teacher's criticism vs. without feedback) × 2 (high PP vs. low PP) factorial experiment. The first condition was identical to Study 1. In contrast, the second condition (PP levels) was manipulated by creating a warm and supportive environment for the participants, as [70] suggested. When the high PP group participants arrived at the laboratory, they were greeted warmly by the research assistants by their names with smiles on their faces. The participants were also asked to express genuine interest in their well-being and their concerns about the experiment or courses, which were listened to attentively. Afterward, participants of the high PP group were instructed to have mindfulness practices such as meditation and deep breathing (see [71] for reference) for five minutes, while their counterparts in the low PP group did not receive these treatments. The participants of the low PP group sat in front of the computer and watched the scene of April's Breakdown of the movie Revolutionary Road–for five minutes. ChatGPT suggested this clip to be able to cause one's feelings of depression, anxiety, loneliness, or general negativity. Participants' alpha brainwave was detected and recorded as an indicator of PP after the onset of the training course.

## Participants

To calculate the sample size for a 3 × 2 mixed analysis of variance (ANOVA) using G*Power 3.1.9.7, the following values in the *F tests* ANOVA were applied for the repeated measures between factors A priori: effect size = 0.4, alpha error = 0.05, power = 0.80, number of groups = 6. The minimum suggested sample size was 80, and Study 2 recruited 102 college-level EFL learners ($n = 102$, $M_{age} = 22.3$, $SD_{age} = 2.1$, Male = 48, Female = 54) through social media. They were randomly and evenly assigned into six groups (17 for each group).

## Research procedure

The research procedure of Study 2 was similar to that of Study 1; however, Study 2 used a 3 × 2 between-subject factorial design. The participants would be randomly assigned to one of the six groups. While the teacher's positive feedback was provided during course teaching, the participants' PP was primed before the experiment; in other words, the participants were given an English reading comprehension task, which would be identical to Study 1. While working on the tasks, the EEG data were collected to detect brainwave oscillations, given that alpha brainwave was considered to measure one's neural efficiency. All participants would need to complete two sessions of the reading courses, and the experiment lasted about 30 minutes. Study 2 took place between November 2023 and May 2024.

## Statistical analysis

To properly examine the main and interaction effects of the two independent variables in Study 2, a two-way ANOVA was performed to assess the statistical significance of the independent variables (i.e., with/without the teacher's positive feedback or criticism and participants' PP) and the relationship with the dependent variable (i.e., participants' neural efficiency). The residuals of the dependent variable show minimal skewness (0.248) and kurtosis (-0.261), which are well within the acceptable thresholds for normality. These results indicate that the residuals are approximately normally distributed, satisfying the normality assumption of ANOVA.

## Results and discussion of Study 2

The results of two-way ANOVA are shown in Table 3. The participants' PP significantly affected their neural efficiency (i.e., alpha brainwave), $F (1) = 37.963$, $p = .000$, $\eta_p^2 = .283$, which supported H2a. Conversely, the teacher's positive feedback also displayed a significant

**Table 3. Results of two-way ANOVA.**

| Source | Type III Sum of Squares | df | Mean Square | F | Sig. | $\eta_p^2$ |
|---|---|---|---|---|---|---|
| Corrected model | 15.288[a] | 5 | 3.058 | 11.687 | .000 | .378 |
| Intercept | 189.525 | 1 | 189.525 | 724.437 | .000 | .883 |
| PP | 9.932 | 1 | 9.932 | 37.963 | .000 | .283 |
| PFB | 2.160 | 2 | 1.080 | 4.128 | .019 | .079 |
| PP * PFB | 3.453 | 2 | 1.726 | 6.599 | .002 | .121 |
| Error | 25.115 | 96 | .262 | | | |
| Total | 232.560 | 102 | | | | |
| Corrected Total | 40.403 | 101 | | | | |

[a]. R Squared = .387 (Adjusted R Squared = .364)

Dependent Variable: Neural Efficiency (Alpha Brainwave)

effect ($F_{(2)}$ = 5.324, $p$ = .019, $\eta_p^2$ = .079); hence, H2b was supported. The interaction of these two variables significantly affected neural efficiency, indicated as $F_{(2)}$ = 6.599, $p$ = .00, $\eta_p^2$ = .121 (please see Table 3). These findings provided valuable information regarding the effectiveness of the teacher's positive feedback on neural efficiency in English classes and supported H2. Furthermore, this study provided empirical evidence that EFL learners' PP benefits their neural efficiency.

Based on the results of Study 1, Study 2 was designed to extend our current understanding of the role of EFL learners' PP and instructors' positive feedback on EFL learners' neural efficiency. Study 2 echoed the arguments of [72, 73], who suggested that emphasizing positive psychological factors may enhance EFL learners' well-being and academic success. Moreover, the results of Study 2 were in line with the findings of [74], who asserted that teachers' support contributes significantly to EFL learners' positive emotions, impacting their overall language learning experience. Furthermore, as facilitators in learning English, teachers play a crucial role in shaping students' positive emotions [75]. Such a finding extended the research of [76] and further suggested that support from the teacher would have a beneficial effect on learners' neural efficiency.

Additionally, EFL teachers should optimize learners' neural pathways for language acquisition by providing positive feedback. Study 2 explored this issue from the perspective of EFL learners' neural efficiency using a neuroscientific approach, which has yet to be attempted in previous studies to the best of current knowledge. However, direct exploration of the relationship between EFL learners' neural efficiency and their learning outcomes still needs to be explored. Therefore, Study 3 was conducted.

## Study 3

Studies 1 and 2 focused on whether the teacher's positive feedback activates EFL learners' PP and the effects of these two variables on neural efficiency. However, the association between EFL learners' neural efficiency and their learning outcomes remains unknown [77]. Study 3 used a single-factor between-subjects design to fill this research gap. The independent variable in Study 3 was the participants' neural efficiency, which was still measured using the alpha brainwave power band. The dependent variable was learning outcomes, assessed using the participants' learning content (The difference in scores between the post-test and the pre-test). The assessments included multiple-choice and short-answer questions; participants received scores on a scale of 100. Pre- and post-test questions came from TOEFL reading tests with similar difficulty levels as the course content but with different topics. All these questions were reviewed by three experienced EFL teachers (with more than ten years of teaching experience) to ensure the appropriateness and suitability to be used in Study 3.

### Participants

Following the rationale of Studies 1 and 2, the sample size of Study 3 was decided based on the results of G*Power 3.1.9.7 with parameters of effect size ρ (= .40), statistical power (= .80), and alpha level (= .05) for the correlation analysis. The suggested minimum number of participants was 44. Accordingly, Study 3 attempted to recruit more than 44 participants, and 54 participants ($n$ = 54, $M_{age}$ = 21.9, $SD_{age}$ = 1.1, Male = 27, Female = 27) completed the experiment.

### Research procedure

After obtaining written consent from the participants, they proceeded with the pre-test. Prior to commencing the experiment, the participants were directed to position the EEG electrodes and sit with their eyes shut. Soothing music was utilized to stabilize the participants' emotional

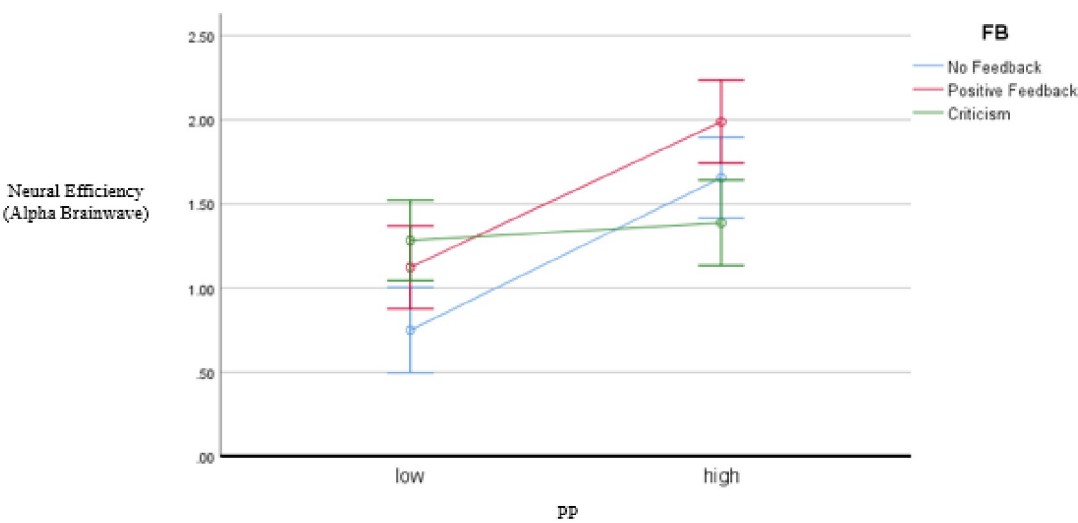

**Fig 6. The results of two-way ANOVA analysis of Study 2.**

state for 2 minutes, during which their EEG readings were captured and saved once the experiment commenced. Positioned 1 meter away from a computer screen, the participants viewed the online course for 30 minutes. Following this session, a five-minute intermission was given before the post-test was conducted. The entire process of Study 3 took approximately one hour for each participant to complete. Study 3 was carried out between January 2024 and July 2024. A Pearson Correlation analysis was performed to examine the association between two variables, i.e., EFL learners' neural efficiency and their learning outcome.

## Results and discussions of Study 3

The information in Table 3 and Fig 6 indicated that the relationship between neural efficiency and learning outcomes was significant and robust ($r = .563$, $p = .000$) [78], supporting H3 (please see Table 4 and Fig 7 below).

Research indicates that neural efficiency is a critical factor in language learning. It has been asserted that individuals with highly efficient neural processing can quickly understand grammar, vocabulary, and pronunciation during learning [79]. The findings of Study 3 indicated that traits of neural efficiency coordinate neural resources, which has been suggested as a defining characteristic of proficient reading [80]. Moreover, the results of Study 3 extended the postulation of [81], who argued that at the cerebral cortical level, elite performers in EFL learning exhibit high neural efficiency by reducing neural effort within the constraints of the imposed task demand.

**Table 4. Results of Pearson Correlation coefficient between EFL learners' neural efficiency and learning outcomes.**

|  | Neural Efficiency | Learning Outcomes |
|---|---|---|
| Neural Efficiency |  |  |
| Learning Outcomes | $r = .563^{***}$ |  |

$^{***}p < .001$

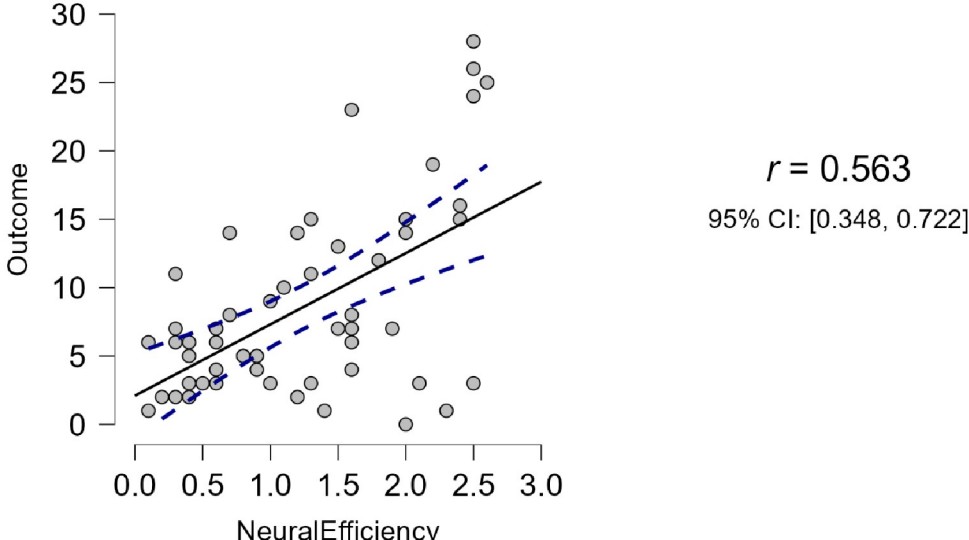

**Fig 7. Correlation of the participants' neural efficiency and learning outcomes.**

## General discussions, limitations, and implications

### General discussion

PP has significantly influenced educational practices since the early 21st century, highlighting the benefits of positive emotions like gratitude, kindness, optimism, zest, and joy on students' well-being and academic performance [82]. Recognizing PP's role in education, it has been applied to EFL learning to enhance students' welfare and success in learning English [83]. However, more research is needed on incorporating PP interventions into language teaching and learning [84]. This study addresses this gap by conducting three experiments to explore the neural mechanisms in EFL learners' PP, focusing on the impact of teachers' positive feedback and neural efficiency.

Studies 1 and 2 found positive teacher feedback significantly improved EFL learners' PP. Moreover, PP positively affected neural efficiency, and the interaction between PP and positive feedback significantly enhanced neural efficiency. Positive feedback combined with learners' PP promotes a positive attitude, self-confidence, and resilience, leading to better educational outcomes. Learners with strong PP are more prepared to face challenges and stay committed to their studies, achieving academic goals even without the teacher's feedback [4].

Moreover, large effect sizes observed in the impact of PP interventions (e.g., positive feedback) indicate meaningful practical improvements. For example, even small improvements in learners' emotional well-being can increase motivation and persistence, essential for long-term success in language learning. Learners experiencing enhanced well-being are likelier to engage with challenging tasks and maintain focus during complex activities, such as preparing for high-stakes exams like TOEFL or IELTS. The positive psychological gains observed in this study—such as reduced anxiety, increased confidence, and greater emotional resilience—contribute to learners' ability to communicate effectively in real-world scenarios, such as presentations, interviews, or daily conversations in English.

From the neuroscientific perspective, evidence from this present research indicates that the alpha band of EEG suppression during task performance, observed in learners with higher emotional well-being, reflects reduced cognitive effort and improved neural efficiency. This

suggests that when learners experience positive emotions, they are more likely to engage with language tasks in a relaxed state, allowing cognitive processes to become more automatic and efficient.

Study 3's findings align with the statement of [85], who show that neural efficiency varies with different interactional contexts. Previous research [86] indicates that higher cognitive abilities are linked to reduced brain activity, suggesting better neural efficiency. Individuals with higher cognitive abilities use fewer brain resources [87]. Our study demonstrated that PP, combined with positive feedback and neural efficiency, improved EFL learners' cognitive abilities and learning outcomes. [80] suggested that superior inhibitory control, facilitated by neural efficiency, leads to more effective use of the language network during reading, explaining the link between neural efficiency and learning outcomes in EFL learners. Accordingly, it is important for EFL teachers to leverage PP-based strategies to enhance learners' emotional well-being, which in turn supports neural efficiency. For example, uplifting feedback, peer encouragement, and intrinsic motivation through personalized learning goals can reduce learners' cognitive load, promoting more efficient learning. Case in point, this finding supports the integration of PP strategies, such as consistent positive reinforcement, goal-setting, and autonomy support, into language instruction to boost learners' emotional engagement and motivation over time, leading to tangible improvements in their language proficiency.

## Limitations

This study encountered several limitations. All three studies were conducted as laboratory experiments, allowing controlled examination of variable interactions and causality. However, this approach needs ecological validity. Future research should investigate the effects of PP interventions in real-world classroom settings where environmental variables (e.g., peer interaction, task complexity) may influence learners' emotional and cognitive engagement differently from lab conditions. Secondly, despite calculating sample sizes based on effect size, statistical power, and alpha values, the findings' generalizability could be improved due to the small sample sizes. Larger-scale surveys are recommended to enhance generalizability.

Moreover, examining the long-term impact of positive feedback strategies on language proficiency could offer deeper insights into how sustained interventions shape learning outcomes over time. Equally important, since learner responses to feedback may vary across different cultural contexts, future research should explore the effectiveness of positive feedback strategies (e.g., autonomy support vs. praise) across different EFL environments, such as in Western and Asian educational systems. This approach would offer cross-cultural validation of the study's findings and identify culturally sensitive teaching practices.

Additionally, while the study designs suggested a correlation between EFL learners' PP and their learning achievements, direct evidence is still lacking. Lastly, individual differences such as personality traits (e.g., openness contributing to positive emotions and personal growth, as noted by [88]), cognitive abilities, working memory capacity, and prior language exposure influence PP and neural efficiency [89, 90]. Further investigation into these factors is necessary. It is also important to note that while PP is associated with increases in FAA, they are not necessarily the same [31], and future research is advised to develop a direct approach to measure one's PP.

Feedback types could have various effects. Verbal feedback (e.g., spoken praise during lessons) may have a more immediate emotional impact, fostering a sense of connection between teachers and learners and helping to build confidence in oral communication tasks. Conversely, written feedback (e.g., comments on assignments or online messages) provides learners a permanent reference to revisit, potentially reinforcing positive emotions over time and

supporting self-directed learning. It may be especially beneficial in asynchronous online learning environments, where learners process feedback at their own pace. Therefore, future research should compare the effectiveness of these feedback types in different learning contexts, such as face-to-face and online classes, to determine which methods are most effective for enhancing emotional well-being and improving language outcomes.

While innovative in combining neuroscientific and behavioral methods, the study's experimental design is not without potential confounding variables that could influence the results. One such confound is the individual differences in learners' baseline PP and motivation levels. Some students may naturally have higher levels of psychological well-being or intrinsic motivation, which could affect how they respond to feedback. Additionally, language proficiency and prior exposure to English could vary among participants, influencing their cognitive load and neural efficiency during learning tasks. Cultural differences in how feedback is perceived could also play a role, as learners from certain cultures may respond differently to positive reinforcement.

Another confound could arise from teacher-student rapport; participants with better relationships with instructors may perceive feedback more positively, which affects their performance. Moreover, fatigue or attention levels during EEG sessions could impact neural measurements, as participants with higher fatigue might exhibit reduced neural efficiency, regardless of feedback. To mitigate these potential confounds, the study employed random assignment and controlled for participants' proficiency levels. However, future research could further address these issues by using larger, more diverse samples and longitudinal designs to monitor changes over time.

## Theoretical implications

The originality of this study lies in its three-experiment design to explore whether the teacher's positive feedback enhances EFL learners' PP, thereby improving neural efficiency and learning outcomes. These findings have significant theoretical implications, confirming the role of the teacher's positive feedback in cultivating EFL learners' PP from a neuroscientific perspective. The study discovered a possible effect of EFL learners' PP on the relationship between the teacher's positive feedback and neural efficiency. The interaction between PP and positive feedback significantly affected neural efficiency, highlighting their importance in EFL learning, as neural efficiency is strongly associated with positive learning outcomes. The empirical evidence provides new insights into cultivating and using PP in EFL learning. This research establishes a new pathway to understand EFL learners' PP through a neuroscientific foundation, with follow-up research potentially expanding knowledge and contributions in this area.

## Practical implications

The findings of this study suggest that online EFL instructors can enhance learners' PP by incorporating positive reinforcement strategies and fostering learner autonomy, helping to alleviate students' fear of failure [91]. Improved PP is linked to more efficient neural processing and can positively impact learning outcomes. However, educators need to recognize that positive feedback alone is insufficient to enhance neural efficiency—it must be paired with perceptual processing tasks (e.g., lexical decision or timed reading exercises) that engage students cognitively.

This study highlights that PP plays a pivotal role in learners' neural efficiency, reinforcing that cognitive and emotional support must be integrated into teaching methods. Supporting findings of [74], our research underscores the importance of meaningful feedback in shaping learners' emotional experiences. Constructive and uplifting feedback promotes language

acquisition and learners' well-being, fostering a positive learning environment. By integrating these research-based approaches into teacher training programs, educators can develop more effective pedagogical strategies and provide personalized support to learners. The study's insights can also inform the development of innovative language learning technologies, such as adaptive platforms that offer tailored feedback and support. This research contributes to a deeper understanding of optimizing language learning experiences in the digital age by exploring the broader implications for online learning contexts and feedback mechanisms.

The implications of these findings may also extend beyond Taiwan and are relevant to EFL instructors globally, particularly in online and non-immersion settings. Educators can balance positive feedback with structured cognitive challenges to build learners' confidence while developing their language proficiency. Moreover, these strategies are adaptable to different educational systems, helping teachers create effective learning environments across various cultural contexts.

## Conclusions

Although the significance of the teacher's positive feedback and learners' PP in EFL learning has been extensively researched, the mechanisms through which positive feedback promotes PP and how PP leads to better learning outcomes require more empirical evidence. This research advanced understanding by designing three experiments to explore these issues, aiding EFL education scholars and practitioners in understanding how positive feedback fosters or enhances learners' PP in the EFL context. Additionally, the study examined how PP and positive feedback improve EFL learners' neural efficiency. The results demonstrated that the teacher's positive feedback effectively enhanced PP. The interaction between positive feedback and PP significantly influenced neural efficiency. Furthermore, a significant positive association was found between neural efficiency and learning outcomes, supporting the neural efficiency hypothesis in EFL learning. Understanding the complex connection between neural efficiency and EFL learning outcomes allows for customizing instructional approaches to enhance neural functions and support language learners.

## Supporting information

**S1 Data.**
(XLSX)

## Acknowledgments

I sincerely appreciate the editor and reviewers' constructive feedback and insightful suggestions, which greatly enhanced the quality of this work. I deeply value their time and effort in reviewing the manuscript.

## Author Contributions

**Data curation:** Liwei Hsu.

**Formal analysis:** Liwei Hsu.

**Methodology:** Liwei Hsu.

**Project administration:** Liwei Hsu.

**Resources:** Liwei Hsu.

**Writing – original draft:** Liwei Hsu.

**Writing – review & editing:** Liwei Hsu.

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
