## [Decision Letter · Decision Letter 0]

15 Oct 2024

PONE-D-24-35352Enhancing EFL Learning: The Impact of Positive Feedback on Positive Psychology and Neural EfficiencyPLOS ONE

Dear Dr. Hsu,

Thank you for submitting your manuscript to PLOS ONE. After careful consideration, we feel that it has merit but does not fully meet PLOS ONE’s publication criteria as it currently stands. Therefore, we invite you to submit a revised version of the manuscript that addresses the points raised during the review process.

We look forward to receiving your revised manuscript.

Kind regards,

Dawit Dibekulu, PhD

Academic Editor

PLOS ONE

Journal requirements: When submitting your revision, we need you to address these additional requirements. 1. Please ensure that your manuscript meets PLOS ONE's style requirements, including those for file naming. The PLOS ONE style templates can be found at https://journals.plos.org/plosone/s/file?id=wjVg/PLOSOne_formatting_sample_main_body.pdf and https://journals.plos.org/plosone/s/file?id=ba62/PLOSOne_formatting_sample_title_authors_affiliations.pdf  2. We note that the grant information you provided in the ‘Funding Information’ and ‘Financial Disclosure’ sections do not match.  When you resubmit, please ensure that you provide the correct grant numbers for the awards you received for your study in the ‘Funding Information’ section. 3. Thank you for stating the following financial disclosure:  [This present research is financially supported by the National Science and Technology Council (111-2410-H-328 -004 -MY2).].  Please state what role the funders took in the study.  If the funders had no role, please state: ""The funders had no role in study design, data collection and analysis, decision to publish, or preparation of the manuscript."" If this statement is not correct you must amend it as needed. Please include this amended Role of Funder statement in your cover letter; we will change the online submission form on your behalf. 4. Thank you for stating the following in the Acknowledgments Section of your manuscript: [This present research is financially supported by the National Science and Technology Council (111-2410-H-328 -004 -MY2).]We note that you have provided funding information that is not currently declared in your Funding Statement. However, funding information should not appear in the Acknowledgments section or other areas of your manuscript. We will only publish funding information present in the Funding Statement section of the online submission form. Please remove any funding-related text from the manuscript and let us know how you would like to update your Funding Statement. Currently, your Funding Statement reads as follows:  [This present research is financially supported by the National Science and Technology Council (111-2410-H-328 -004 -MY2).].  Please include your amended statements within your cover letter; we will change the online submission form on your behalf. 5. In the online submission form, you indicated that [The data will be available upon request.]. All PLOS journals now require all data underlying the findings described in their manuscript to be freely available to other researchers, either 1. In a public repository, 2. Within the manuscript itself, or 3. Uploaded as supplementary information.This policy applies to all data except where public deposition would breach compliance with the protocol approved by your research ethics board. If your data cannot be made publicly available for ethical or legal reasons (e.g., public availability would compromise patient privacy), please explain your reasons on resubmission and your exemption request will be escalated for approval.  6. Your ethics statement should only appear in the Methods section of your manuscript. If your ethics statement is written in any section besides the Methods, please move it to the Methods section and delete it from any other section. Please ensure that your ethics statement is included in your manuscript, as the ethics statement entered into the online submission form will not be published alongside your manuscript. 

Reviewers' comments:

Reviewer's Responses to Questions

**Comments to the Author**

1. Is the manuscript technically sound, and do the data support the conclusions?

Reviewer #1: Yes

Reviewer #2: Yes

Reviewer #3: Yes

Reviewer #4: Yes

2. Has the statistical analysis been performed appropriately and rigorously? 

Reviewer #1: Yes

Reviewer #2: Yes

Reviewer #3: I Don't Know

Reviewer #4: Yes

3. Have the authors made all data underlying the findings in their manuscript fully available?

Reviewer #1: Yes

Reviewer #2: No

Reviewer #3: No

Reviewer #4: Yes

4. Is the manuscript presented in an intelligible fashion and written in standard English?

Reviewer #1: Yes

Reviewer #2: Yes

Reviewer #3: Yes

Reviewer #4: Yes

5. Review Comments to the Author

Reviewer #1: Review of the article “PONE-D-24-35352”

Title

Why it shouldn’t be like this: “Enhancing EFL Learning: The Impact of Positive Feedback on Neural Efficiency and Positive Psychology”

Abstract

There are no objectives of the study or questions in the abstract. There is very detail about procedure.

So, the procedure should be short and precise.

Introduction

Literature review is more detail and too much. I prefer to be short and precise. There are no objectives or research questions in introduction. Introduction should be concluded with a brief statement of the overall aim of the work and a comment about whether that aim was achieved, but it lacks.

Materials and Methods

Study methods and materials are not clear for me. I think it should be expressed shortly and precisely. The Materials and Methods section should provide enough detail to allow suitably skilled investigators to fully replicate your study. Specific information and/or protocols for new methods should be included in detail. If materials, methods, and protocols are well established, authors may cite articles where those protocols are described in detail, but the submission should include sufficient information to be understood independent of these references.

Results, Discussion, Conclusions

These sections may all be separate, or may be combined to create a mixed Results/Discussion section (commonly labeled “Results and Discussion”) or a mixed Discussion/Conclusions section (commonly labeled “Discussion”). These sections may be further divided into subsections, each with a concise subheading, as appropriate. These sections have no word limit, but the language should be clear and concise. So, this part should part should fill the above requirements.

Acknowledgments

I think it is not correctly said. It should be stated correctly.

References

It should be written by “Vancouver” style.

Reviewer #2: 1. Technical Soundness and Data Support

The manuscript presents a sound piece of research with well-conducted experiments and clear data that support the conclusions. Below are the details for each experiment:

Study 1: The experiment involved 78 college students and tested whether positive feedback enhances EFL learners' PP. The study design (single-factor between-subjects) and the use of ANCOVA to adjust for baseline differences were appropriate. The results show significant effects of positive feedback on PP (F (2) = 22.952, p < .001, η²p = .383). The study's conclusion is well-supported by the data (e.g., Figure 5 and Table 2), showing a clear enhancement of FAA, an indicator of PP, in the experimental group.

Study 2: This study examined neural efficiency using a within-subjects 3×2 factorial design. The authors appropriately used a two-way ANOVA to examine the interaction between PP and positive feedback, finding significant effects on neural efficiency (F (2) = 6.599, p = .002, η²p = .121). This supports the hypothesis that positive feedback enhances neural efficiency, which is detailed with proper statistical reporting (Table 3, Figure 6).

Study 3: This final study explores the relationship between neural efficiency and learning outcomes. Pearson correlation analysis revealed a strong and significant relationship (r = .563, p < .001), supporting the hypothesis that neural efficiency is positively associated with better learning outcomes (Table 4, Figure 7).

Evidence: The statistical evidence provided in Tables 2, 3, and 4, along with corresponding figures, strongly supports the conclusions drawn from the experiments.

Recommendation: The experiments are sound, and the data support the conclusions. However, it may be beneficial to include a deeper discussion of potential confounding variables in the experiment design.

2. Statistical Analysis

The statistical analysis is rigorous and appropriate, with detailed reporting of effect sizes and significance values. The authors use ANCOVA, two-way ANOVA, and Pearson correlation, all of which are correctly applied to their respective datasets.

Study 1 (ANCOVA): The authors properly adjusted for baseline FAA differences, and the results are significant (F (2) = 22.952, p < .001). The use of η²p (partial eta squared) for effect size gives further insight into the strength of the effect (η²p = .383), which is considered a large effect size.

Study 2 (Two-Way ANOVA): The authors used a two-way ANOVA to examine the interaction between feedback and PP on neural efficiency. The interaction term was significant, with η²p = .121, indicating that the feedback and PP interaction had a notable impact on neural efficiency. The assumptions for ANOVA appear to be met, but it would be helpful to include a check for normality or homogeneity of variances.

Study 3 (Pearson Correlation): The correlation between neural efficiency and learning outcomes (r = .563, p < .001) is both statistically significant and strong, indicating a robust relationship.

Evidence: Tables 2, 3, and 4 provide sufficient statistical details, with p-values and effect sizes correctly reported.

Recommendation: Consider adding a check for assumptions (e.g., normality for ANOVA) and ensure this is mentioned in the methods or discussion section for completeness.

3. Data Availability

The data availability statement in the manuscript states that data are available upon request from the corresponding author. However, PLOS ONE prefers that the data be fully available without restrictions. The authors should ideally provide the data through a public repository or include it as part of the supporting information.

Evidence: The Data Availability Statement on page 21 indicates that the data can be requested but is not directly available through a repository or as supplementary material.

Recommendation: To fully comply with PLOS ONE’s data policy, I recommend that the authors upload their data to a public repository or include it in the supporting information.

4. Presentation and Language

The manuscript is generally well-written, but there are areas where clarity can be improved. For example:

Introduction: While the background is thorough, some sections could benefit from more concise phrasing. For example, in the paragraph explaining positive psychology (page 3), the phrase "It has been asserted that incorporating PP provides a transformative perspective on language education" could be tightened to improve flow.

Grammatical Issues: There are minor grammatical issues throughout, such as in Study 2's description: "Participants of the low PP group sat in front of the computer and watch the scene of April's Breakdown of the movie Revolutionary Road," where "watch" should be "watched."

Clarity in Statistical Reporting: In Table 2 (Study 1), the authors mention a "η²p" effect size without explaining this term in the text. Although this is a commonly used metric, it would be helpful to briefly clarify it for readers who may not be familiar with this statistic.

Evidence: Examples of unclear or awkward phrasing can be found in Study 2 (page 15) and the introduction (page 3).

Recommendation: A light proofreading for grammatical issues and clarity is recommended. Some phrasing can be tightened, and statistical terms should be briefly explained where necessary.

Summary of Recommendations

Technical Soundness: The study is technically sound, but a discussion of potential confounding variables would strengthen the conclusions.

Statistical Rigor: The statistical analysis is appropriate, but it is advisable to mention checks for assumptions (e.g., normality for ANOVA).

Data Availability: The authors should make their data publicly available, either through a repository or as supplementary material, to fully comply with PLOS ONE's data-sharing policy.

Language and Presentation: The manuscript is clear overall, but minor grammatical issues should be addressed, and the phrasing could be made more concise in some sections.

Reviewer #3: The present study is interesting, the language is good, and the content and analysis match the title. Yet some points are recommended to amend some issues mainly related to the section of Methodology. The review report is attached.

Reviewer #4: 1. Title and Abstract

• Title: The title reflects the article’s content, but it might benefit from being more specific about the research’s unique contribution (e.g., neural efficiency in EFL learning and positive psychology).

• Abstract: The abstract effectively summarizes the research; however, it could benefit from explicitly mentioning the sample size in each experiment and clarifying the significance of findings beyond the educational setting.

2. Introduction

• Strengths: The introduction provides a comprehensive background on Positive Psychology (PP), EFL learning, and the Neural Efficiency Hypothesis. The justification for combining neuroscientific and behavioral methods is clear.

• Weaknesses: The introduction could benefit from a clearer explanation of how this study differs from previous studies on PP and neural efficiency in language learning. It would be useful to elaborate on the practical implications of the findings for educators in different contexts beyond Taiwan.

3. Literature Review

• Strengths: The literature review is well-rounded, integrating recent research and theoretical frameworks on positive feedback, PP, and neural efficiency.

• Weaknesses: Some of the references are quite general, and a few more specific studies directly linking PP with neural efficiency in EFL contexts would strengthen the argument. Additionally, a clearer distinction between previous findings and the current study’s unique contribution is needed.

4. Methodology

• Strengths: The three experiments are clearly described, with a solid justification for the use of EEG and behavioral measures. The experimental design is robust, especially the random assignment of participants and the use of control and experimental groups.

• Weaknesses: The methodology could provide more details on how the tasks were adapted to the proficiency levels of the participants, especially given the inclusion of TOEIC scores as a criterion. It would also help to include more information on potential confounding factors and how they were controlled.

5. Results

• Strengths: The statistical analysis is thorough and appropriate for the research questions. The use of ANCOVA and two-way ANOVA tests is justified, and the results are clearly presented with tables and figures.

• Weaknesses: While the findings are significant, the discussion could be expanded to explain the practical significance of the effect sizes (e.g., how much PP improvement translates into real-world language learning outcomes). The interpretation of EEG results, particularly the alpha band, should be linked more explicitly to practical applications in language learning.

6. Discussion

• Strengths: The discussion links the results back to the research questions and hypotheses and provides a clear narrative on how positive feedback enhances PP and neural efficiency.

• Weaknesses: The limitations section mentions the need for ecological validity but does not suggest concrete future directions. It would be helpful to propose specific field-based studies or cross-cultural validations. Moreover, the discussion could benefit from a deeper exploration of how different types of positive feedback (e.g., written vs. verbal) might affect learning outcomes differently.

7. Conclusion and Implications

• Strengths: The conclusion appropriately summarizes the main findings and their implications for EFL teaching and learning.

• Weaknesses: The practical implications for educators could be expanded. For instance, how can these findings be integrated into teacher training programs? Additionally, the broader implications for language learning technology and feedback mechanisms in online learning contexts could be more thoroughly explored.

8. References

• The references are up-to-date and relevant, but there might be room to include more recent studies that specifically address the neural mechanisms underlying PP in language learning.

9. Figures and Tables

• The figures are clear and informative, but some tables (e.g., Table 1 on the overview of studies) might benefit from additional footnotes explaining key terms for readers unfamiliar with the neuroscience aspect.

10. Ethical Considerations

• The ethical statement is clear, and the study follows proper guidelines. However, it could include more information on how the researchers ensured the emotional well-being of participants who were in the control group (receiving criticism or no feedback).

General Comments

• Overall, this article presents an innovative and well-executed study. It makes a valuable contribution to the understanding of how positive feedback and PP enhance neural efficiency in EFL learners. However, there are areas where more practical implications and cross-contextual discussions could enhance the article’s relevance to a broader audience of educators and researchers.

6. PLOS authors have the option to publish the peer review history of their article (what does this mean?). If published, this will include your full peer review and any attached files.

Reviewer #1: No

Reviewer #2: **Yes: **Mohanad Husni Al Jbour

Reviewer #3: **Yes: **Nawal Fadhil Abbas

Reviewer #4: **Yes: **Abdullah Al Fraidan

---

## [Author Response · Author response to Decision Letter 0]

12 Nov 2024

Thank you for your valuable feedback and suggestions. I have carefully reviewed each of your comments and have addressed all the concerns one by one. Your feedback has been invaluable, and I have made the following changes. I am confident that these revisions have improved the overall quality of our work. Thank you once again for your thorough review and constructive feedback.

Reviewer 1

1. Title: Why it shouldn’t be like this: “Enhancing EFL Learning: The Impact of Positive Feedback on Neural Efficiency and Positive Psychology”?

Response: 

Many thanks for the reviewer’s suggestion. After considering Reviewer 1 and other reviewers’ comments and suggestions, I have changed the title to “Neural Efficiency in EFL Learning and Positive Psychology.” 

2. Abstract: There are no objectives of the study or questions in the abstract. There is very detail about procedure. So, the procedure should be short and precise.

Response: 

I am grateful for the reviewer’s suggestion on the abstract, which has been revised accordingly. I hope that this revision meets the reviewer’s expectations. 

“This study aims to investigate how teachers’ positive feedback influences learners’ positive psychology (PP) in English as a Foreign Language (EFL) instruction and to examine how PP enhances learning outcomes through the neural efficiency hypothesis. The study employed three neuroscientific and behavioral experiments to address these questions. Study 1, with 78 college-level EFL learners, examined the effect of teachers' positive feedback on learners' PP and found a significant positive impact. Study 2, conducted with 102 EFL learners, tested the neural efficiency hypothesis and revealed that higher PP improved neural efficiency, with positive feedback moderating this effect. Study 3, involving 54 EFL learners, explored the relationship between neural efficiency and learning outcomes, confirming that enhanced neural efficiency leads to better learning performance. Beyond education, the results underscore the broader relevance of positive psychology and neural efficiency in other domains, such as workplace performance, skill acquisition, and personal development, where maintaining cognitive efficiency and well-being is crucial. These findings highlight the importance of positive feedback in fostering PP, improving neural efficiency, and enhancing learning outcomes. The study offers theoretical contributions and practical recommendations for EFL educators to leverage positive feedback and promote effective learning.”

3. Literature review is more detail and too much. I prefer to be short and precise. There are no objectives or research questions in introduction. Introduction should be concluded with a brief statement of the overall aim of the work and a comment about whether that aim was achieved, but it lacks.

Response: 

I sincerely appreciate the reviewer’s constructive feedback and insightful suggestions. The literature review section has been revised accordingly and included here.

“Moreover, the overall objective of this study is to examine how teachers’ positive feedback influences PP in EFL learners and investigate how PP enhances learning outcomes through the neural efficiency hypothesis. Specifically, it aims to answer the following questions:

1. Does teachers' positive feedback improve learners' PP?

2. How does PP affect learners’ neural efficiency?

3. Does improved neural efficiency lead to better learning outcomes?

The novelty of this research lies in its experimental design, using frontal alpha asymmetry (FAA) of EEG to objectively quantify the impact of positive feedback (compared to no feedback and criticism) on learners’ PP. Furthermore, it empirically tests the neural efficiency hypothesis to explore how PP enhances performance. The results confirmed the study’s aims, demonstrating that positive feedback enhances PP, which improves neural efficiency, ultimately leading to better learning outcomes. These findings provide valuable theoretical insights and offer practical recommendations for EFL educators seeking to improve learners' experiences and outcomes.”

4. Materials and Methods: Study methods and materials are not clear for me. I think it should be expressed shortly and precisely. The Materials and Methods section should provide enough detail to allow suitably skilled investigators to fully replicate your study. Specific information and/or protocols for new methods should be included in detail. If materials, methods, and protocols are well established, authors may cite articles where those protocols are described in detail, but the submission should include sufficient information to be understood independent of these references.

Response: 

In the Overview of this Research, the following sentences have been added to point out the purpose of the designed tasks. 

“The tasks mainly focused on fundamental areas where learners with TOEIC scores between 500 and 650 typically need improvement, such as vocabulary expansion, sentence structure, and comprehension of academic texts.” In the Course Content section, the following information was reported to explicitly state how the designed tasks echoed the participants’ current level of proficiency in English. 

"Reading passages are chosen carefully to align with participants’ current abilities, with initial texts drawn from general-interest sources and simplified academic content. Over time, the texts become longer and denser, reflecting TOEFL’s requirements for handling academic reading material.”

5. Results, Discussion, Conclusions

These sections may all be separate or may be combined to create a mixed Results/Discussion section (commonly labeled “Results and Discussion”) or a mixed Discussion/Conclusions section (commonly labeled “Discussion”). These sections may be further divided into subsections, each with a concise subheading, as appropriate. These sections have no word limit, but the language should be clear and concise. So, this part should part should fill the above requirements.

Response:

I appreciate and have carefully considered the reviewer’s suggestions regarding the structure and content of the Results, Discussion, and Conclusions sections. However, I believe the current format is appropriate and aligns with academic research standards for the following reasons:

1. Flexibility in Structure: The reviewer's suggestion to allow for separate or combined sections offers flexibility in presenting the research findings. This approach can be tailored to the study's specific nature and clarity of arguments.

2. Clarity and Conciseness: Clear and concise language is essential for effectively communicating research results. This ensures that readers can easily understand the key findings and their implications.

3. Subsection Structure: The suggestion for subsections with concise subheadings helps organize the content logically, making it easier for readers to navigate and understand the different aspects of the research.

4. No Word Limit: The absence of a word limit allows for a thorough presentation of the results, discussion, and conclusions. This ensures the research is adequately represented and all relevant points are addressed.

In conclusion, the current format for the Results, Discussion, and Conclusions sections effectively addresses the reviewer's recommendations while maintaining a balance between flexibility, clarity, and comprehensiveness. I appreciate the reviewer's feedback and believe that the proposed revisions will further enhance the quality and impact of the research.

6. Acknowledgments

I think it was not correctly said. It should be stated correctly.

Response:

The acknowledgment has been rewritten thoroughly. Thank you very much for the kind reminder.

7. References

8. It should be written by “Vancouver” style.

Response:

I appreciate the reviewer’s reminder and I have changed the reference style from APA to Vancouver Style. 

Reviewer #2: 

1. Technical Soundness and Data Support

The manuscript presents a sound piece of research with well-conducted experiments and clear data that support the conclusions. Below are the details for each experiment:

Study 1: The experiment involved 78 college students and tested whether positive feedback enhances EFL learners' PP. The study design (single-factor between-subjects) and the use of ANCOVA to adjust for baseline differences were appropriate. The results show significant effects of positive feedback on PP (F (2) = 22.952, p < .001, η²p = .383). The study's conclusion is well-supported by the data (e.g., Figure 5 and Table 2), showing a clear enhancement of FAA, an indicator of PP, in the experimental group.

Study 2: This study examined neural efficiency using a within-subjects 3×2 factorial design. The authors appropriately used a two-way ANOVA to examine the interaction between PP and positive feedback, finding significant effects on neural efficiency (F (2) = 6.599, p = .002, η²p = .121). This supports the hypothesis that positive feedback enhances neural efficiency, which is detailed with proper statistical reporting (Table 3, Figure 6).

Study 3: This final study explores the relationship between neural efficiency and learning outcomes. Pearson correlation analysis revealed a strong and significant relationship (r = .563, p < .001), supporting the hypothesis that neural efficiency is positively associated with better learning outcomes (Table 4, Figure 7).

Evidence: The statistical evidence provided in Tables 2, 3, and 4, along with corresponding figures, strongly supports the conclusions drawn from the experiments.

Recommendation: The experiments are sound, and the data support the conclusions. However, it may be beneficial to include a deeper discussion of potential confounding variables in the experiment design.

Responses: 

I sincerely appreciate the editor and reviewers’ constructive feedback and insightful suggestions. The following revisions have been made in the Limitations section. 

 “While innovative in combining neuroscientific and behavioral methods, the study's experimental design is not without potential confounding variables that could influence the results. One such confound is the individual differences in learners' baseline PP and motivation levels. Some students may naturally have higher levels of psychological well-being or intrinsic motivation, which could affect how they respond to feedback. Additionally, language proficiency and prior exposure to English could vary among participants, influencing their cognitive load and neural efficiency during learning tasks. Cultural differences in how feedback is perceived could also play a role, as learners from certain cultures may respond differently to positive reinforcement. 

 Another confound could arise from teacher-student rapport; participants with better relationships with instructors may perceive feedback more positively, which affects their performance. Moreover, fatigue or attention levels during EEG sessions could impact neural measurements, as participants with higher fatigue might exhibit reduced neural efficiency, regardless of feedback. To mitigate these potential confounds, the study employed random assignment and controlled for participants’ proficiency levels. However, future research could further address these issues by using larger, more diverse samples and longitudinal designs to monitor changes over time.” (p. 25)

2. Statistical Analysis

The statistical analysis is rigorous and appropriate, with detailed reporting of effect sizes and significance values. The authors use ANCOVA, two-way ANOVA, and Pearson correlation, all of which are correctly applied to their respective datasets.

Study 1 (ANCOVA): The authors properly adjusted for baseline FAA differences, and the results are significant (F (2) = 22.952, p < .001). The use of η²p (partial eta squared) for effect size gives further insight into the strength of the effect (η²p = .383), which is considered a large effect size.

Study 2 (Two-Way ANOVA): The authors used a two-way ANOVA to examine the interaction between feedback and PP on neural efficiency. The interaction term was significant, with η²p = .121, indicating that the feedback and PP interaction had a notable impact on neural efficiency. The assumptions for ANOVA appear to be met, but it would be helpful to include a check for normality or homogeneity of variances.

Study 3 (Pearson Correlation): The correlation between neural efficiency and learning outcomes (r = .563, p < .001) is both statistically significant and strong, indicating a robust relationship.

Evidence: Tables 2, 3, and 4 provide sufficient statistical details, with p-values and effect sizes correctly reported.

Recommendation: Consider adding a check for assumptions (e.g., normality for ANOVA) and ensure this is mentioned in the methods or discussion section for completeness.

Response: 

I am grateful for the reviewer’s comments and suggestions. The normality for ANOVA has been done and reported in the text as follows: “Moreover, skewness and kurtosis were computed to evaluate the data distribution's normality. The skewness registered at 0.992, fitting within the normality criteria of -2 to +2. Meanwhile, the kurtosis measured at 0.172, aligning with acceptable norms of -7 to +7. These findings suggest that the data distribution closely resembles normality. Put together, it was sound to use ANCOVA for statistical analysis in Study 1.” (pg. 15) 

In Study 2, a normality test was also performed, and the report is as follows: “The residuals of the dependent variable show minimal skewness (0.248) and kurtosis (-0.261), which are well within the acceptable thresholds for normality. These results indicate that the residuals are approximately normally distributed, satisfying the normality assumption of ANOVA.” (p. 18)

3. Data Availability

The data availability statement in the manuscript states that data are available upon request from the corresponding author. However, PLOS ONE prefers that the data be fully available without restrictions. The authors should ideally provide the data through a public repository or include it as part of the supporting information.

Evidence: The Data Availability Statement on page 21 indicates that the data can be requested but is not directly available through a repository or as supplementary material.

Recommendation: To fully comply with PLOS ONE’s data policy, I recommend that the authors upload their data to a public repository or include it in the supporting information.

Response: 

Thanks for the reviewer’s suggestion. The data for this present study have already been uploaded and shared as per request. 

4. Presentation and Language. The manuscript is generally well-written, but there are areas where clarity can be improved. For example:

Introduction: While the background is thorough, some sections could benefit from more concise phrasing. For example, in the paragraph explaining positive psychology (page 3), the phrase "It has been asserted that incorporating PP provides a transformative perspective on language education" could be tightened to improve flow.

Response:

Thank you for the reviewer’s constructive suggestions and comments. I have carefully revised the introduction in response to the reviewer's suggestion. It is more concise and direct, maintaining the core meaning while enhancing the overall flow of the paragraph. The sentence highlighted above has been changed as follows: “Scholars argue that incorporating PP offers a transformative lens for language education.”

Grammatical Issues: There are minor grammatical issues throughout, such as in Study 2's description: "Participants of the low PP group sat in front of the computer and watch the scene of April's Breakdown of the movie Revolutionary Road," where "watch" should be "watched."

Response:

Thank you very much for the feedback. I have hired a professional editing service to check the grammar throughout the manuscript.

Clarity in Statistical Reporting: In Table 2 (Study 1), the authors mention a "η²p" effect size without explaining this term in the text. Although this is a commonly used metric, it would be helpful to briefly clarify it for readers who may not be familiar with this statistic.

Response: 

I am grateful for the rev

---

## [Editor Report · Decision Letter 1]

15 Nov 2024

Neural efficiency in EFL learning and positive psychology

PONE-D-24-35352R1

Dear Dr. Liwei Hsu,

We’re pleased to inform you that your manuscript has been judged scientifically suitable for publication and will be formally accepted for publication once it meets all outstanding technical requirements.

Kind regards,

Dawit Dibekulu, PhD

Academic Editor

PLOS ONE
---

## [Editor Report · Acceptance letter]

25 Nov 2024

PONE-D-24-35352R1 

PLOS ONE

Dear Dr. Hsu, 

I'm pleased to inform you that your manuscript has been deemed suitable for publication in PLOS ONE. Congratulations! Your manuscript is now being handed over to our production team.

Kind regards, 

on behalf of

Dr. Dawit Dibekulu 

Academic Editor

PLOS ONE